# Looking at the Bigger Picture—Considering the Hurdles in the Struggle against Nanoplastic Pollution

**DOI:** 10.3390/nano11102536

**Published:** 2021-09-28

**Authors:** Sophie M. Briffa

**Affiliations:** Department of Metallurgy and Materials Engineering, University of Malta, MSD2080 Msida, Malta; sophie.briffa@um.edu.mt

**Keywords:** nanoplastics, environmental ageing, nanoplastic toxicity

## Abstract

Plastics are considered one of the most serious environmental global concerns as they are ubiquitous and contribute to the build-up of pollution. In August 2020, the BBC reported that scientists found 12–21 million tonnes of tiny plastic fragments floating in the Atlantic Ocean. After release into the environment, plastics from consumer items, such as cosmetics and biomedical products, are subject to degradation and break down into microplastics (<5 mm in diameter) and eventually into nanoplastics (<100 nm in at least one dimension). Given their global abundance and environmental persistence, exposure of humans and animals to these micro- and nano- plastics is unavoidable. “We urgently need to know more about the health impact of microplastics because they are everywhere”, says Dr Maria Neira, Director at the World Health Organization. Nanoplastics are also an emerging environmental concern as little is known about their generation, degradation, transformation, ageing, and transportation. Owing to their small size, nanoplastics can be trapped by filter-feeding organisms and can enter the food chain at an early stage. Therefore, there is a gap in the knowledge that vitally needs to be addressed. This minireview considers how nanoplastic research can be made more quantifiable through traceable and trackable plastic particles and more environmentally realistic by considering the changes over time. It considers how nanoplastic research can use industrially realistic samples and be more impactful by incorporating the ecological impact.

## 1. Introduction

Plastics are considered one of the most serious environmental issues and global concerns. They contribute to most of the build-up of pollution and are now found far and wide in marine, freshwater, and terrestrial systems [1,2,3]. Plastics have been observed in some of the Earth’s most remote regions, such as the Mariana Trench [4]. After being released into the environment, plastics from consumer items such as cosmetics, paints, textiles, biomedical products, pharmaceuticals, and cleaning products are subject to degradation (e.g., physical, chemical, and biological weathering), causing them to break down into microplastics (<5 mm in diameter) and eventually into nanoplastics (<100 nm in at least one of their dimensions) [1,2,5]. In August 2020, the BBC reported that scientists found 12–21 million tonnes of tiny plastic fragments floating in the Atlantic Ocean [6]. Microplastics have been found to have possible toxic effects on interaction with biota [7]. The full health impact of these micro- and nano- plastics is yet to be fully understood. Given their global abundance and environmental persistence, exposure of humans and animals to these micro- and nano- plastics is unavoidable [5]. Recently, microplastics have been found in table salt, commercially cultured mussels, fish, beer, bottled water, and tap water [8]. Nanoplastics can also enter places far more remote than microplastics (i.e., cross cellular barriers and other barriers e.g., blood–brain barrier) [9]. Both micro- and nano- plastics have been detected in human tissues [10].

Tackling the plastic pollution problem is a global priority. In 2017, the European Commission confirmed it would focus on plastics production and, work towards achieving a strategy for plastics in a circular economy by 2030. Plastic pollution is also linked to the European Green Deal, which provides an action plan to boost the efficient use of resources by moving to a clean, circular economy, restoring biodiversity, and cutting pollution. Furthermore, the UN resolution on Marine Litter and Microplastics and the UN’s 2030 Agenda for Sustainable Development included pursuing clean water and sanitation (SDG6), enabling responsible consumption and production (SDG12), and safeguarding life below water (SDG14) and life on land (SDG15). These goals aim to achieve a better and more sustainable future for all.

However, to tackle the issue of plastic pollution, research into ways of achieving this is first needed. As Dr Maria Neira, Director, Department of Public Health, Environment and Social Determinants of Health, at the World Health Organisation, says, “We urgently need to know more about the health impact of microplastics because they are everywhere” [11]. This cry for research into the impacts of microplastics has been echoed by the UK Department for Business, Energy and Industrial Strategy, who in 2019 highlight that the Government appreciates that there are knowledge and evidence gaps around the risks of microplastics on ecosystems and humans. There is therefore a need to steer the scientific community to focus research on closing the key knowledge gaps to understand the real effects these materials are exerting on ecosystems and humans.

Steering the scientific community efforts towards closing these gaps in the knowledge involves understanding the gaps, as well as the progress in the field of nanoplastic research and complementary fields, such as characterisation instrumentation. Research is necessary to understand the realistic effects of nanoplastics, through a structured understanding of the physicochemical properties of the nanoplastics found in the environment, their fate, behaviour, and toxicity. Furthermore, it involves the ability to replicate these scenarios for the purposes of research studies (Figure 1).

In this short article, issues relating to nanoplastic traceability, the realism of laboratory testing samples, environmental and lifecycle changes, and risk assessment have been considered. By highlighting these issues from literature, light can be shed on the considerations that must made in this research field.

## 2. Considering the Hurdles in the Struggle against Nanoplastic Pollution

### 2.1. Rendering Research Quantifiable through Traceable and Trackable Plastic Particles

One issue related to nanoplastic research is the difficulty encountered to identify microplastics of various sizes, shapes, and polymer types fully and reliably from complex environmental matrices using a single analytical method [10]. To date, combinations of physical and chemical analyses are used [10]. The most commonly used techniques are microscopy, Fourier transform infrared spectroscopy (FTIR), Raman spectroscopy, pyrolysis-gas chromatography-mass spectrometry (Pyro-GC-MS), and X-ray fluorescence (XRF) [3]. Few of these techniques are portable and able to allow for actual environmental analysis. Furthermore, as the size of micro-plastics decreases, it becomes increasingly time-consuming to identify them [10]. In fact, nanoplastic particles have proven challenging to separate and characterise using traditional techniques. The quantification and identification of micro- and nano- plastics remain labour and cost intensive and still require the achievement of high-sample throughput analysis. This is due to the lack of analytical methods that can be used to characterise these small particles at low environmentally realistic concentrations affordably. There is a need to improve and develop new methods and hyphenate existing methods to reduce the identification time and effort and detect sub-micron plastics in environmental samples [10].

Whilst there is the need to improve the capabilities of analytical techniques, there are other ways to deal with this issue in the interim. To combat this issue, researchers have begun looking into staining and doping micro- and nanoplastics to render them detectable. Nile Red adsorbed onto plastic surfaces, for example, renders them fluorescent when irradiated with blue light [12]. Different types of plastic displayed different fluorescent colours [12]. Other approaches include using micro- and nano- plastics doped with metal/metal–oxide materials [13,14]. These are easily measurable with existent analytical techniques such as inductively coupled plasma-mass spectrometry (ICP-MS) that can characterise small particles at low concentrations. It should be ensured that the chosen dopant does not instil ecotoxicity if any leeching occurs. Furthermore, leeching should be avoided as that would render the nanoplastic no longer detectable in the manner it was prepared for.

Besides considering the efficient and effective attachment of a dopant to a nanoplastic, one should consider the influence that the dopant has on the properties of the plastic. Changes in properties could, in turn, influence the behaviour and toxicity profiles of these nanoplastics. This may be acceptable for early-stage preliminary lab studies; yet, for more advanced and environmentally realistic studies that focus on tracking nanoplastics throughout their lifetime, this will cause issues.

The environmental nanosafety community has extensively investigated the toxicity effects of metal and metal–oxide nanomaterials. This vital information can be extracted to the field of micro- and nano- plastics to ensure that dopants that are more easily detected are not considered harmful. These more easily detectable small plastics will be traceable and trackable through the environment. Hence, these will allow for a better understanding of the environmental behaviour, fate, and toxicity of micro- and nano- plastics.

### 2.2. Rendering Research Samples Industrially Realistic through an Understanding of the Physicochemical Properties

It has been reported that the health impact of microplastic ingestion on exposed organisms depends on the nature and size of the particles [7]. Polystyrene particles with different sizes and surface modifications have previously been shown to have different negative effects on wildlife [15]. These findings indicate that the choice of plastic samples for research studies must take into consideration realistic physicochemical properties, hence the characteristics of micro- and nano- plastics found in the environment.

Plastic particles in the environment are rarely ever perfectly spherical, despite spherical microbeads being used as representative plastics in the majority of studies [16]. The reason for the widespread use of spherical particles is that they are generally used for calibrating instruments and are therefore easier to analyse. In fact, the majority of toxicity studies focus on the use of polystyrene micro- and nano- beads, despite polyethylene and polypropylene fragments being the most common in aquatic environments [7,17]. Micro- and nano- plastics are a product of weathering and breakdown mechanisms. These small plastics tend to be irregularly shaped fragments comprised of an array of different shapes and sizes. Researchers have started to investigate the fragmentation processes of macro- and micro- plastics [18,19,20,21,22]. This information should be used to guide the choice of plastic particles for future studies.

Therefore, research should look to steer its studies towards the use of fragmented plastics rather than spherical plastic particles. Studies should consider using a top–down approach (such as dissolution, ball milling, grinding, laser ablation, or cutting) to prepare plastic fragments from commercially available plastic pellets, as these methods will help replicate environmental weathering processes. This would, in turn, guarantee that plastics are more realistic and similar to what is present in the environment.

It is important to note that there is a huge variety of micro- and nano- plastics, not only with respect to variations in physical properties such as size and shape, but also with respect to chemical composition. This poses another considerable challenge to risk assessors [23]. The chemical aspect has been considered more extensively in risk assessment than the physical aspect has. Since bulk plastic pollution has now been a problem for several years, the research community is well aware of the chemical composition of the majority of plastics found in the environment. Hence, research focuses on the most commonly used plastics such as polyethylene and polystyrene and takes into consideration additives such as plasticizers.

Despite this, commercial micro- and nano- beads being used in research have surfactants at their surface to reduce the influence of aggregation. These surfactants may not necessarily be present on plastic fragments found in the environment. Moreover, the surfactants may enhance or modify their interaction pathway with aquatic organisms. Only a few studies have reported protocols to prepare model micro- and nano- plastics [7].

One way of further ensuring chemical consistency between samples used for risk assessment research studies and those found in the environment is by using industrially and commercially available plastics. This certifies that the chemistry of the research materials matches that of plastic particles that end up in the environment.

### 2.3. Rendering Research Studies Environmentally Realistic by Considering the Changes over Time

Nanoplastics are an emerging concern in the environment as little is known about their generation, degradation, transformation, transportation, and toxicity in the environment [24]. Once micro- and nano- plastic fragments are formed, they can continue to undergo changes during their lifetime. An understanding of the environmental fate of nanoplastics is essential for risk assessments, but it is complicated by the fact that chemical and physical properties such as shape and size change over time (“ageing”). Assessing transformation processes over time provides critical information on nanoplastics’ persistence and hence environmental accumulation. Ageing has been found to induce very complex physicochemical changes in polymeric materials, depending on the type of polymer and environmental conditions [25]. The formation of these particles starts due to changes occurring over time; namely, weathering, which is considered a very slow process. In the marine environment, abiotic degradation through sunlight, oxidants, and physical stress is generally recognised as a starting point of plastic degradation [25]. Ageing can affect polymer composition, physical integrity, surface properties, and degradation. Hence, it gives a new identity to the particles that may affect their physical, chemical, and biological environmental responses. These changes can happen through various complex processes, including photo-oxidation, variations in temperature, hydrolysis, mechanical abrasion, swelling, the release of additives, biodegradation, protein corona formation, pollutant adsorption, and colonisation by microorganisms. Knowing how microplastic particles weather is important for understanding the ecological impacts of the most common type of marine debris [19]. For instance, Vroom et al. found that ageing of microplastics promoted their ingestion by marine zooplankton [26].

Currently, although the ageing of plastics has been studied, it is often overlooked for risk assessment studies, and micro- and nano- plastics are studied in their original form. In fact, Liu et al. point out that the information regarding the impact of ageing on the environmental behaviour of microplastics is still lacking [27]. To obtain a more accurate assessment of the potential behaviour of these plastics, further research based on changes these plastics may undergo during their lifecycle is needed.

For instance, photodegradation may lead to transformation products with higher reported toxicity. Liu et al. found that ageing by UV or O_3_ exposure, the mobility and contaminant-mobilizing ability of spherical polystyrene nanoplastics (hydrodynamic diameter = 487.3 ± 18.3 nm) was drastically enhanced in saturated loamy sand [28]. Furthermore, consideration should be made for all details of real-life conditions, such as water salinity, mixed microbiological populations, and natural cycles of temperature and light [25].

Besides environmental factors such as light, water, and mechanical stress, the formation of a protein corona and biofilms also accounts for the ageing of plastic materials [25]. Proteins can associate with biopolymers, to form a protein corona associated with the particle, and continuously exchange with proteins from the surrounding environment; hence, continuously giving a new biological identity to the particles, which may affect their biological responses. When the surrounding environment of the particles also contains microorganisms, biofilms are formed. These should also be considered.

Finally, one of the most significant parameters when it comes to ageing is time. Ageing studies of metal–oxide nanomaterials have shown that changes do not occur linearly as a function of time or temperature, and transformations are far more complex than anticipated [29]. In their work, Brandon et al. found that experimental weathering for microplastics was also more complex than predicted. The chemical bonds did not change linearly with time, and there was variability in weathering between the combinations of plastic, weathering experiment, and bond type measured [19]. These are important considerations to make when determining study durations. Longer-term studies are more realistic of environmental conditions; however, studies are often limited by project duration and deadlines. It is important to find a balance to achieving longer-term realistic studies within project duration limitations. One way of overcoming this is to attempt to mimic results from real-life scenarios over shorter periods of time, hence accelerating ageing. Further research is still needed to determine how best this can be achieved. As ageing studies are so dependent on time, the data collected from these studies is very valuable. This data will be invaluable to feed into predictive models for the design of safer plastics with the aim to move towards a circular economy.

### 2.4. Rendering Research Impactful by Reviewing the Ecological Impact

Due to their size, microplastics have been shown to be ingested by a broad range of organisms, particularly by organisms at the lower end of the food chain, subsequently transferring to top consumers by feeding [12,30]. This could have threatening effects on ecosystems and human health as they have been shown to cause tissue damage or death [30].

The risks that microplastics pose to marine life and humans are widely recognised and have been included in national and international marine protection strategies, policies, and legislation [12]. It has been demonstrated in the laboratory that, at high exposure concentrations and under specific circumstances, microplastics can induce physical and chemical toxicity [31]. The exact extent of the hazardous nature of nanoplastics is continuously being debated, and despite their ubiquitous presence, there is a general scarcity of data regarding their uptake and toxicity [2]. There is a correlation of increased toxicity of metals/metal–oxides from microscale to nanoscale. Hence, scientific consensus is that nanoplastics will be more toxic than microplastics, but evidence to support this is scarce, especially regarding environmentally relevant nanoplastics. Continuing to study and monitor their ecological impact will be instrumental to promote new environmental legislation.

There are many uncertainties as to whether ‘classical’ risk assessment approaches are sufficient because of long-term accumulation and exposure [23]. In fact, Kelpsiene et al. found that lifetime exposure (103 days) of a common freshwater invertebrate, *Daphnia Magna*, exposed to sub-lethal concentrations of polystyrene nanoparticles showed different results to short-term exposures [30].

To carry out appropriate risk assessment, as well as analytical methods for the detection and quantification of plastic particles, adequate ecotoxicological methods are needed to determine the exposure levels in the environment and food chain [23]. ‘Classical’ risk assessment is designed for dissolved chemicals and not for particulate matters [23]. Furthermore, to date, ecological studies have focused on pristine spherical micro- and nano- plastics; these studies, though important to understand the materials’ behaviour, lack an environmentally realistic aspect.

## 3. Conclusions

Tackling the problem of plastic pollution is linked to developing a sustainable future and relates to global strategies, such as those of the UN and EU. However, pollution by micro- and nano- plastics is complex, and managing it effectively requires multidimensional responses. There is a need to steer the scientific community to focus research on understanding the real effects these materials are exerting on ecosystems and humans. Researchers need to find different approaches for detecting and generating microplastics and nanoscale plastics to mimic those found in the environment more closely. Future studies need to understand how aged micro- and nano- plastic particle fragments may affect biota through lifetime studies replicating environmental concentrations to provide information for predictions of future scenarios. This will guarantee a sustainable approach to tackling one of the most significant environmental, social, and economic challenges of the 21st Century.

## Figures and Tables

**Figure 1 nanomaterials-11-02536-f001:**
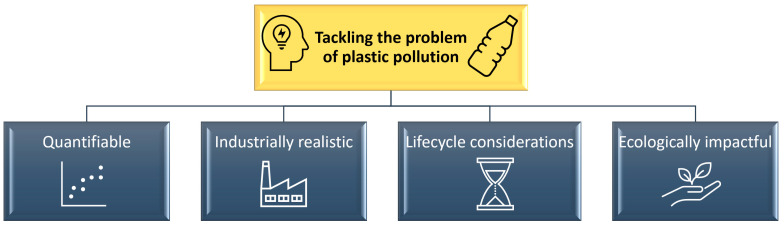
Overlooked aspects that need to be considered when tackling the problem of nanoplastic pollution, a key environmental global concern.

## Data Availability

Not applicable.

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
