# Peer review of "Looking at the Bigger Picture—Considering the Hurdles in the Struggle against Nanoplastic Pollution"

_nanomaterials, 2021, doi:10.3390/nano11102536_

Round 1

Reviewer 1 Report

The minireview by Sophie M Briffa treats the main aspects which can be considered when micro- and nano-plastics should be detected. It is a challange addresed to researchers to solve the pollution problem associated to nanoplastics, as a global priority. The manuscript highlights that alternative routes like Nire Red and the dopant of nanoplastic can be used to reduce the identification time of nanoplastics. To be industrially realistic, irregularly shape fragments should be characterized instead of spherical plastic particles. The factors accontable for aging of nanomaterials as well as the toxicological methods are needed to determine the environmental and ecological impact of nanoplastics. In my opinion I recommend the publishing of paper, as an important issue for the pollution with micro and nanoplastics.

Author Response

Thank you for your comments and the realisation of such a vital topic.

Reviewer 2 Report

I revised the mini review by Briffa SM titled “Looking at the bigger picture - Considering the hurdles in the struggle against nanoplastic pollution”.

I found the reading very interesting because the Author attempts to face a very challenging topics of the environmental pollution. Even if short and simple, the contribution gives serval hot points of discussions concerning the limitation of the current investigations and approaches used for risk assessment of nanoplastic pollution. The Aurthor suggests also possible strategies to overcame these failures, in order to obtain more realistic and reliable information from the research performed to date and aimed to understand the risk that nanoplastics represent for the health.

Overall the review is complete and well written. I found a possible mistake at the line 227 , “sparce….” perhaps the Author intended “scarce…”

Author Response

Thank you for your comments and the realisation of such a vital topic. On line 227 'sparce' has been changed to 'scarce' as suggested.

Reviewer 3 Report

Very well work. 

Author Response

Thank you for your comments. An updated version has been attached.
